# Contrastive Intrinsic Control for Unsupervised Reinforcement Learning

**Michael Laskin**
UC Berkeley
mlaskin@berkeley.edu

**Hao Liu**
UC Berkeley

**Xue Bin Peng**
UC Berkeley

**Denis Yarats**
NYU, Meta AI

**Aravind Rajeswaran**
UC Berkeley, Meta AI

**Pieter Abbeel**
UC Berkeley, Covariant

## Abstract

We introduce Contrastive Intrinsic Control (CIC), an unsupervised reinforcement learning (RL) algorithm that maximizes the mutual information between state-transitions and latent skill vectors. CIC utilizes contrastive learning between state-transitions and skills vectors to learn behaviour embeddings and maximizes the entropy of these embeddings as an intrinsic reward to encourage behavioural diversity. We evaluate our algorithm on the Unsupervised RL Benchmark (URLB) in the asymptotic state-based setting, which consists of a long reward-free pre-training phase followed by a short adaptation phase to downstream tasks with extrinsic rewards. We find that CIC improves over prior exploration algorithms in terms of adaptation efficiency to downstream tasks on state-based URLB. [1]

Deep RL is a powerful approach toward solving complex control tasks in the presence of extrinsic rewards. Successful applications include playing video games from pixels [1], mastering the game of Go [2, 3], robotic locomotion [4, 5, 6] and dexterous manipulation [7, 8, 9] policies. While effective, the above advances produced agents that are unable to generalize to new downstream tasks beyond the one they were trained to solve. Humans and animals on the other hand are able to acquire skills with minimal supervision and apply them to solve a variety of downstream tasks. In this work, we seek to train agents that acquire skills without supervision with generalization capabilities by efficiently adapting these skills to downstream tasks.

Over the last few years, unsupervised RL has emerged as a promising framework for developing RL agents that can generalize to new tasks. In the unsupervised RL setting, agents are first pre-trained with self-supervised intrinsic rewards and then finetuned to downstream tasks with extrinsic rewards. Unsupervised RL algorithms broadly fall into three categories - knowledge-based, data-based, and competence-based methods[2]. Knowledge-based methods maximize the error or uncertainty of a predictive model [12, 13, 14]. Data-based methods maximize the entropy of the agent's visitation [15, 16]. Competence-based methods learn skills that generate diverse behaviors [17, 18]. This work falls into the latter category of competence-based exploration methods.

Unlike knowledge-based and data-based algorithms, competence-based algorithms simultaneously address both the exploration challenge as well as distilling the generated experience in the form of reusable skills. This makes them particularly appealing, since the resulting skill-based policies (or skills themselves) can be finetuned to efficiently solve downstream tasks. While there are many self-supervised objectives that can be utilized, our work falls into a family of methods that learns skills by maximizing the mutual information between visited states and latent skill vectors. Many earlier

---

[1]Project website and code: `https://sites.google.com/view/cicneurips2022/`
[2]These categories for exploration algorithms were introduced by [10] and inspired by [11].

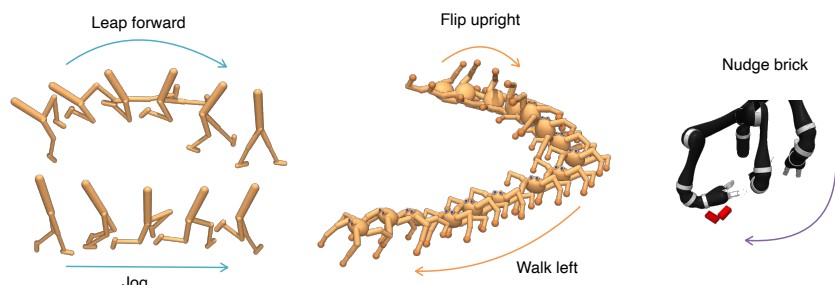

Figure 1: Qualitative visualizations of unsupervised skills discovered in Walker, Quadruped, and Jaco arm environments. The Walker learns to balance and move, the Quadruped learns to flip upright and walk, and the 6 DOF robotic arm learns how to move without locking. Unlike prior competence-based methods for continuous control which evaluate on OpenAI Gym (e.g. [17]), which reset the environment when the agent loses balance, CIC is able to learn skills in fixed episode length environments which are much harder to explore (see Appendix I).

works have investigated optimizing such objectives [17, 18, 19, 20]. However, competence-based methods have been empirically challenging to train and have under-performed when compared to knowledge and data-based methods [21].

In this work, we take a closer look at the challenges of pre-training agents with competence-based algorithms. We introduce Contrastive Intrinsic Control (CIC) – an exploration algorithm that uses a new estimator for the mutual information objective. CIC combines particle estimation for state entropy [22, 15] and noise contrastive estimation [23] for the conditional entropy which enables it to both generate diverse behaviors *(explore)* and discriminate high-dimensional continuous skills *(exploit)*. To the best of our knowledge, CIC is the first exploration algorithm to utilize noise contrastive estimation to discriminate between state transitions and latent skill vectors. Empirically, we show that CIC adapts to downstream tasks more efficiently than prior exploration approaches on the state-based Unsupervised Reinforcement Learning Benchmark (URLB). CIC achieves 79% higher returns on downstream tasks than prior competence-based algorithms and 18% higher returns than the next-best exploration algorithm overall.

## 1 Background and Notation

**Markov Decision Process:** We operate under the assumption that our system is described by a Markov Decision Process (MDP) [24]. An MDP consists of the tuple $(\mathcal{S}, \mathcal{A}, \mathcal{P}, r, \gamma)$ which has states $s \in \mathcal{S}$, actions $a \in \mathcal{A}$, transition dynamics $p(s'|s, a) \sim \mathcal{P}$, a reward function $r$, and a discount factor $\gamma$. In an MDP, at each timestep $t$, an agent observes the current state $s$, selects an action from a policy $a \sim \pi(\cdot|s)$, and then observes the reward $r$ and next state $s'$ once it acts in the environment. Note that usually $r$ refers to an extrinsic reward. However, in this work we will first be pre-training an agent with intrinsic rewards $r^{\text{int}}$ and finetuning on extrinsic rewards $r^{\text{ext}}$.

For convenience we also introduce the variable $\tau = (s, s')$ which is a tuple denoting a transition between two consecutive states. Importantly, $\tau$ does not denote a state-action trajectory. In addition to the standard MDP notation, we will also be learning skills $z \in \mathcal{Z}$ where $\mathcal{Z}$ is the skill set, which can be a discrete or continuous real-valued vector space, and our policy will be skill-conditioned $a \sim \pi(\cdot|s, z)$.

**Unsupervised Skill Discovery through Mutual Information Maximization:** Most competence-based approaches to exploration maximize the mutual information between states and skills. Our work and a large body of prior research [17, 20, 18, 25, 26, 27] aims to maximize a mutual information objective with the following general form:

$$I(\tau; z) = \mathcal{H}(z) - \mathcal{H}(z|\tau) = \mathcal{H}(\tau) - \mathcal{H}(\tau|z) \tag{1}$$

Competence-based algorithms use different choices for $\tau$ and can condition on additional information such as actions or starting states. For a full summary of competence-based algorithms and their objectives see Table 2.

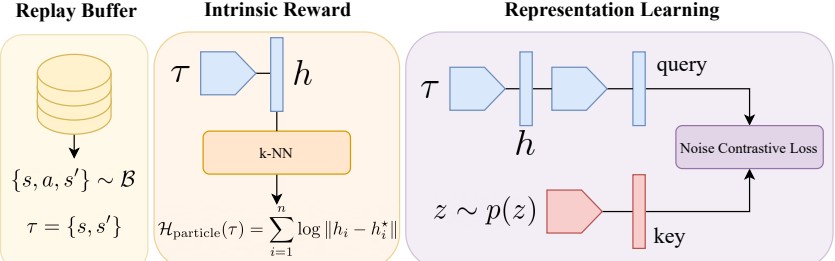

Figure 2: Architecture illustrating the practical implementation of CIC . During a gradient update step, random $\tau = (s, s')$ tuples are sampled from the replay buffer, then a particle estimator is used to compute the entropy and a noise contrastive loss to compute the conditional entropy. The contrastive loss is backpropagated through the entire architecture. The entropy and contrastive terms are then scaled and added to form the intrinsic reward. The RL agent is optimized with a DDPG [29].

**Lower Bound Estimates of Mutual Information:** The mutual information $I(s; z)$ is intractable to compute directly. Since we wish to maximize $I(s; z)$, we can approximate this objective by instead maximizing a lower bound estimate. Most known mutual information maximization algorithms use the variational lower bound introduced in [28]:

$$I(\tau; z) = \mathcal{H}(\tau) - \mathcal{H}(\tau|z) \geq \mathcal{H}(\tau) + \mathbb{E}_{\tau, z}[\log q(\tau|z)] \tag{2}$$

The variational lower bound can be applied to both decompositions of the mutual information. The design decisions of a competence-based algorithm therefore come down to (i) which decomposition of $I(\tau; z)$ to use, (ii) whether to use discrete or continuous skills, (iii) how to estimate $H(z)$ or $H(\tau)$, and finally (iv) how to estimate $H(z|\tau)$ or $H(\tau|z)$.

## 2   Motivation

Results from the recent Unsupervised Reinforcement Learning Benchmark (URLB) [21] show that competence-based approaches underperform relative to knowledge-based and data-based baselines on DeepMind Control (DMC). We argue that the underlying issue with current competence-based algorithms when deployed on harder exploration environments like DMC has to do with the currently used estimators for $I(\tau; z)$ rather than the objective itself. To produce structured skills that lead to diverse behaviors, $I(\tau; z)$ estimators must (i) explicitly encourage diverse behaviors and (ii) have the capacity to discriminate between high-dimensional continuous skills. Current approaches do not satisfy both criteria.

*Competence-base algorithms do not ensure diverse behaviors:* Most of the best known competence-based approaches [17, 18, 25, 26], optimize the first decomposition of the mutual information $\mathcal{H}(z) - \mathcal{H}(z|\tau)$. The issue with this decomposition is that while it ensures diversity of skill vectors it does not ensure diverse behavior from the policy, meaning $\max \mathcal{H}(z)$ does not imply $\max \mathcal{H}(\tau)$. Of course, if $H(z) - \mathcal{H}(z|\tau)$ is maximized and the skill dimension is sufficiently large, then $\mathcal{H}(\tau)$ will also be maximized implicitly. Yet in practice, to learn an accurate discriminator $q(z|\tau)$, the above methods assume skill spaces that are much smaller than the state space (see Table 2), and thus behavioral diversity may not be guaranteed. In contrast, the decomposition $I(\tau; z) = \mathcal{H}(\tau) - \mathcal{H}(\tau|z)$ ensures diverse behaviors through the entropy term $\mathcal{H}(\tau)$. Methods that utilize this decomposition include [27, 20].

*Why it is important to utilize high-dimensional skills:* Once a policy is capable of generating diverse behaviors, it is important that the discriminator can distill these behaviors into distinct skills. If the set of behaviors outnumbers the set of skills, this will result in degenerate skills – when one skill maps to multiple different behaviors. It is therefore important that the discriminator can accommodate continuous skills of sufficiently high dimension. Empirically, the discriminators used in prior work utilize only low-dimensional continuous skill vectors. DIAYN [17] utilized 16 dimensional skills, DADS [20] utilizes continuous skills of dimension $2 - 5$, while APS [27], an algorithm that utilizes successor features [30, 31] for the discriminator, is only capable of learning continuous skills with

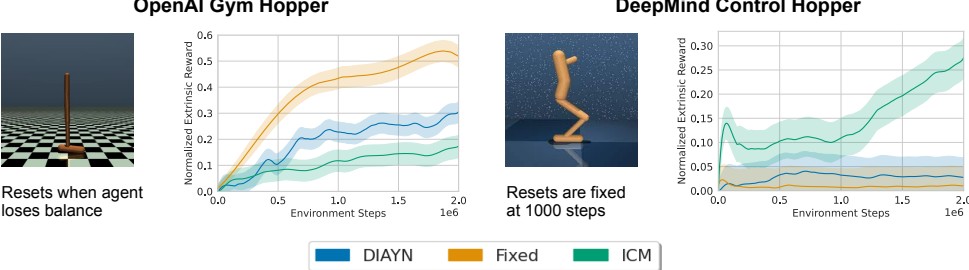

Figure 3: To empirically demonstrate issues inherent to competence-based exploration methods, we run DIAYN [17] and compare it to ICM [12] and a *Fixed* baseline where the agent receives an intrinsic reward of 1.0 for each timestep and no extrinsic reward on both OpenAI Gym *(episode resets when agent loses balance)* and DeepMind Control (DMC) *(episode is fixed for 1k steps)* Hopper environments. Since Gym and DMC rewards are on different scales, we normalize rewards based on the maximum reward achieved by any algorithm ( 1k for Gym,  3 for DMC). While DIAYN is able to achieve higher extrinsic rewards than ICM on Gym, the Fixed intrinsic reward baseline performs best. However, on DMC the Fixed and DIAYN agents achieve near-zero reward while ICM does not. This is consistent with findings of prior work that DIAYN is able to learn diverse behaviors in Gym [17] as well as the observation that DIAYN performs poorly on DMC environments [21]

dimension 10. We show how small skill spaces can lead to ineffective exploration in a simple gridworld setting in Appendix G and evidence that skill dimension affects performance in Fig. 5.

*On the importance of benchmarks for evaluation:* While prior competence-based approaches such as DIAYN [17] were evaluated on OpenAI Gym [32], Gym environment episodes terminate when the agent loses balance thereby leaking some aspects of extrinsic signal to the exploration agent. On the other hand, DMC episodes have fixed length. We show in Fig 3 that this small difference in environments results in large performance differences. Specifically, we find that DIAYN is able to learn diverse skills in Gym but not in DMC, which is consistent with both observations from DIAYN and URLB papers. Due to fixed episode lengths, DMC tasks are harder for reward-free exploration since agents must learn to balance without supervision.

## 3   Method

### 3.1   Contrastive Intrinsic Control

From Section 2 we are motivated to find a lower bound for $I(\tau; z)$ with a discriminator that is capable of supporting high-dimensional continuous skills[3]. Additionally, we wish to increase the diversity of behaviors so that the discriminator can continue learning new skills throughout training. We choose the forward decomposition of MI $I(\tau; z) = \mathcal{H}(\tau) - \mathcal{H}(\tau|z)$ similar to  [27] and estimate the lower bound with Eq. 2. The entropy is estimated $\mathcal{H}(\tau)$ with a particle-based estimator similar to [15]. As such, the primary technical contribution of this work is a novel estimator for the discriminator.

To improve the discriminator, we propose to utilize noise contrastive estimation (NCE) [23] between state-transitions and latent skills as a lower bound for $I(\tau; z)$. It has been shown previously that such estimators provide a valid lower bound for mutual information [33]. However, to the best of our knowledge, this is the first work to investigate contrastive representation learning for intrinsic control.

*Representation Learning:* Specifically, we propose to learn embeddings by parameterizing the discriminator with a contrastive density estimator. This is a novel choice that differs from prior works which utilize a classifier [17] or non-contrastive density estimation [20].

$$\log q(\tau|z) := f(\tau, z) - \log \frac{1}{N} \sum_{j=1}^{N} \exp(f(\tau_j, z)). \tag{3}$$

where $f(\tau, z)$ is any real valued function.

---

[3]In high-dimensional state-action spaces the number of distinct behaviors can be quite large.

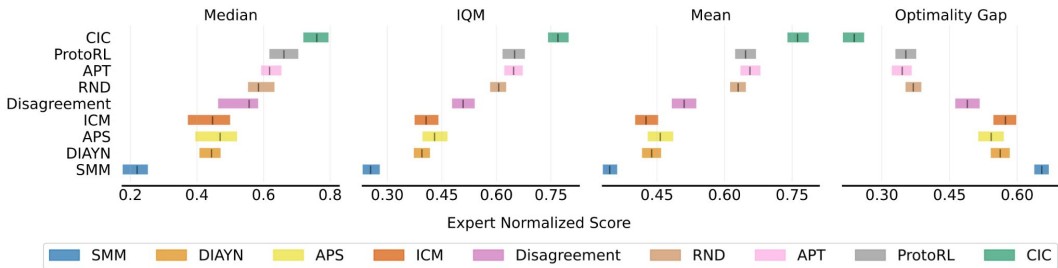

Figure 4: We report the aggregate statistics using stratified bootstrap intervals [35] for 12 downstream tasks on URLB with 10 seeds, so each statistic for each algorithm has 120 seeds in total. We find that overall, CIC achieves leading performance on URLB in terms of the IQM, mean, and OG statistics. As recommended by [35], we use the IQM as our primary performance measure. In terms of IQM, CIC improves upon the next best skill discovery algorithm (APS) by 79% and the next best algorithm overall (ProtoRL) by 18%.

For our practical algorithm, we parameterize this function as $f(\tau, z) = g_{\psi_1}(\tau)^\top g_{\psi_2}(z)/\|g_{\psi_1}(\tau)\|\|g_{\psi_2}(z)\|T$ where $\tau = (s, s')$ is a transition tuple, $g_{\psi_k}$ are neural encoders, and $T$ is a temperature parameter. This inner product is similar to the one used in SimCLR [34].

The representation learning loss backpropagates gradients from the NCE loss which maximizes similarity between state-transitions and corresponding skills.

$$F_{\text{CIC}}(\tau) = \frac{g_{\psi_1}(\tau_i)^\top g_{\psi_2}(z_i)}{\|g_{\psi_1}(\tau_i)\|\|g_{\psi_2}(z_i)\|T} - \log \frac{1}{N} \sum_{j=1}^{N} \exp\left( \frac{g_{\psi_1}(\tau_j)^\top g_{\psi_2}(z_i)}{\|g_{\psi_1}(\tau_j)\|\|g_{\psi_2}(z_i)\|T} \right) \qquad (4)$$

where $N-1$ is the number of negatives. The total number of elements in the summation is $N$ because it includes one positive, so the index $j$ includes the positive index $i$ similar to the objective in [33]. We provide pseudocode for the CIC representation learning loss:

```
"""
PyTorch-like pseudocode for the CIC loss
"""

def cic_loss(s, s_next, z, temp):
    """
    states: s, s_next (B, D), skills: z (B, D)
    """

    tau = concat(s, s_next, dim=1)

    query = query_net(z)
    key = key_net(tau)

    query = normalize(query, dim=1)
    key = normalize(key, dim=1)

    logits = matmul(query, key.T) / temp #logits are (B, B)
    labels = arange(logits.shape[0]) # positives are on diagonal

    # softmax_cross_entropy API same as in PyTorch docs
    loss = softmax_cross_entropy(logits, labels)

    return loss
```

Listing 1: Pseudocode for the CIC loss.

*Intrinsic reward:* Although we have a representation learning objective, we still need to specify the intrinsic reward for the algorithm for which there can be multiple choices. Prior works consider specifying an intrinsic reward that is proportional to state-transition entropy [15], the discriminator [17], a similarity score between states and skills [36], and the uncertainty of the discriminator [37].

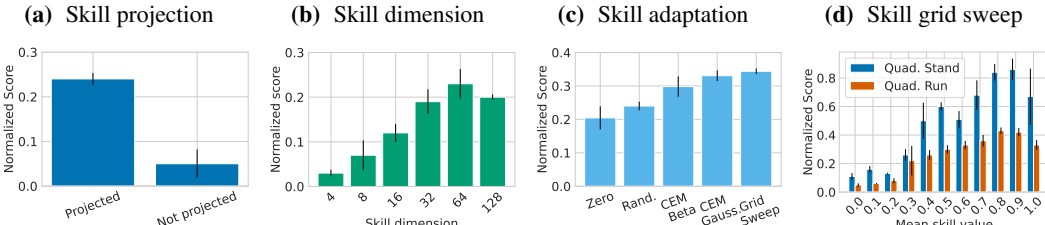

Figure 5: Design choices for pre-training and adapting with skills have significant impact on performance. In (a) and (b) the agent's zero-shot performance is evaluated while sampling skills randomly while in (c) and (d) the agent's performance is evaluated after finetuning the skills vector. *(a)* we show empirically that the projecting skill vectors after sampling them from noise significantly improves the agent's performance. *(b)* The skill dimension is a crucial hyperparameter and, unlike prior methods, CIC scales to large skill vectors achieving optimal performance at 64 dimensional skills. *(c)* We test several adapation strategies and find that a simple grid search performs best given the small 4k step adaptation budget, *(d)* Choosing the right skill vector has substantial impact on performance and grid sweeping allows the agent to select the appropriate skill.

We investigate each of these choices in Section 6 and find that an intrinsic reward that maximizes state-transition entropy coupled with representation learning via the CPC loss defined in Sec. 3.1 is the simplest variant that also performs well (see Table 1), which we use for all other experiments.

For the intrinsic reward, we use a particle estimate [22, 38] as in [15] of the state-transition entropy. Similar to [15, 16] we estimate the entropy up to a proportionality constant, because we want the agent to maximize entropy rather than estimate its exact value.

The APT particle entropy estimate is proportional to the distance between the current visited state transition and previously seen neighboring points.

$$\mathcal{H}_{particle}(\tau) \propto \frac{1}{N_k} \sum_{h_i^\star \in N_k}^{N_k} \log \|h_i - h_i^\star\| \tag{5}$$

where $h_i$ is an embedding of $\tau_i$ shown in Fig. 2, $h_i^*$ is a kNN embedding, $N_k$ is the number of kNNs.

*Explore and Exploit:* With these design choices the two components of the CIC algorithm can be interpreted as *exploration* with intrinsic rewards and *exploitation* using representation learning to distill behaviors into skills. The marginal entropy maximizes the diversity of state-transition embeddings while the contrastive discriminator $\log q(\tau|z)$ encourages exploitation by ensuring that skills $z$ lead to predictable states $\tau$. Together the two terms incentivize the discovery of diverse yet predictable behaviors from the RL agent. While CIC shares a similar intrinsic reward structure to APT [15], we show that the new representation learning loss from the CIC estimator results in substantial performance gains in Sec 6.

## 4   Practical Implementation

Our practical implementation consists of two main components: the RL optimization algorithm and the CIC architecture. For fairness and clarity of comparison, we use the same RL optimization algorithm for our method and all baselines in this work. Since the baselines implemented in URLB [21] use a DDPG[4] [29] as their backbone, we opt for the same DDPG architecture to optimize our method as well (see Appendix B).

*CIC Architecture:* We use a particle estimator as in [15] to estimate $\mathcal{H}(\tau)$. To compute the variational density $q(\tau|z)$, we first sample skills from uniform noise $z \sim p(z)$ where $p(z)$ is the uniform distribution over the $[0, 1]$ interval. We then use two MLP encoders to embed $g_{\psi_1}(\tau)$ and $g_{\psi_2}(z)$,

---

[4]It was recently was shown that a DDPG achieves state-of-the-art performance [39] on DeepMind Control [40] and is more stable than SAC [41] on this benchmark.

and optimize the parameters $\psi_1, \psi_2$ with the CPC loss similar to SimCLR [34] since $f(\tau, z) = g_{\psi_1}(\tau)^T g_{\psi_2}(z)$. We fix the hyperparameters across all domains and downstream tasks. We refer the reader to the Appendices D and E for the full algorithm and a full list of hyperparameters.

*Adapting to downstream tasks:* To adapt to downstream tasks we follow the same procedure for competence-based method adaptation as in URLB [21]. During the first 4k environment interactions we populate the DDPG replay buffer with samples and use the extrinsic rewards collected during this period to finetune the skill vector $z$. While it's common to finetune skills with Cross Entropy Adaptation (CMA), given our limited budget of 4k samples (only 4 episodes) we find that a simple grid sweep of skills over the interval $[0, 1]$ produces the best results (see Fig. 5). After this, we fix the skill $z$ and finetune the DDPG actor-critic parameters against the extrinsic reward for the remaining 96k steps. Note that competence-based methods in URLB also finetune their skills during the first 4k finetuning steps ensuring a fair comparison between the methods. The full adaptation procedure is detailed in Appendix D.

## 5  Experimental Setup

**Environments** We evaluate our approach on tasks from URLB, which consists of twelve downstream tasks across three challenging continuous control domains for exploration algorithms – walker, quadruped, and Jaco arm. Walker requires a biped constrained to a 2D vertical plane to perform locomotion tasks while balancing. Quadruped is more challenging due to a higher-dimensional state-action space and requires a quadruped to in a 3D environment to learn locomotion skills. Jaco arm is a 6-DOF robotic arm with a three-finger gripper to move and manipulate objects without locking. All three environments are challenging in the absence of an extrinsic reward.

**Baselines:** We implemented CIC using the URLB [21] codebase [5] and compare CIC to baselines included in URLB across all three exploration categories. Knowledge-based basedlines include ICM [12], Disagreement [13], and RND [14]. Data-based baselines incude APT [15] and ProtoRL [16]. Competence-based baselines include DIAYN [17], SMM [26], and APS [27]. The closest baselines to CIC are APT, which is similar to CIC but without state-skill CPC representation learning (no discriminator), and APS which uses the same decomposition of mutual information as CIC and also uses a particle entropy estimate for $\mathcal{H}(\tau)$. The main difference between APS and CIC is that APS uses successor features while CIC uses a contrastive estimator for the discriminator. For further details regarding baselines we refer the reader to Appendix C.

**Evaluation:** We follow an identical evaluation to the 2M pre-training setup in URLB. First, we pre-train each RL agent with the intrinsic rewards for 2M steps. Then, we finetune each agent to the downstream task with extrinsic rewards for 100k steps. All baselines were run for 10 seeds per downstream task for each algorithm using the code and hyperparameters provided by URLB [21]. Built on top of URLB, CIC is also run for 10 seeds per task. A total of $1080 = 9$ algorithms $\times$ 12 tasks $\times$ 10 seeds experiments were run for the main results. Importantly, all baselines and CIC use a DDPG agent as their backbone.

To ensure that our evaluation statistics are unbiased we use stratified bootstrap confidence intervals to report aggregate statistics across $M$ runs with $N$ seeds as described in *Rliable* [35] to report statistics for our main results in Fig. 4. Our primary success metric is the interquartile mean (IQM) and the Optimality Gap (OG). IQM discards the top and bottom 25% of runs and then computes the mean. It is less susceptible to outliers than the mean and was shown to be the most reliable statistic for reporting results for RL experiments in [35]. OG measures how far a policy is from optimal (expert) performance. To define expert performance we use the convention in URLB, which is the score achieved by a randomly initialized DDPG after 2M steps of finetuning (20x more steps than our finetuning budget).

## 6  Results

We investigate empirical answers to the following research questions: (Q1) How does CIC adaptation efficiency compare to prior competence-based algorithms and exploration algorithms more broadly?

---

[5]URLB is open-sourced under an MIT license `https://github.com/rll-research/url_benchmark/blob/main/LICENSE`.

(Q2) Which intrinsic reward instantiation of CIC performs best? (Q3) How do the two terms in the CIC objective affect algorithm performance? (Q4) How does skill selection affect the quality of the pre-trained policy? (Q5) Which architecture details matter most?

**Adaptation efficiency of CIC and exploration baslines:** Expert normalized scores of CIC and exploration algorithms from URLB are shown in Fig. 4. We find that CIC substantially outperforms prior competence-based algorithms (DIAYN, SMM, APS) achieving a $79\%$ higher IQM than the next best competence-based method (APS) and, more broadly, achieving a $18\%$ higher IQM than the next best overall baseline (ProtoRL). In further ablations, we find that the contributing factors to CIC's performance are its ability to accommodate substantially larger continuous skill spaces than prior competence-based methods.

**Intrinsic reward specification:** The intrinsic reward for competence-based algorithms can be instantiated in many different ways. Here, we analyze intrinsic reward for CIC with the form $r_{int} = H(\tau) + D(\tau, z)$, where $D$ is some function of $(\tau, z)$. Prior works, select $D$ to be (i) the discriminator [27], (ii) a cosine similarity between embeddings [36], (iii) uncertainty of the discriminator [37], and (iv) just the entropy $D(\tau, z) = 0$ [15]. We run CIC with each of these variants on the walker and quadruped tasks and measure the final mean performance across the downstream tasks (see Tab. 1). The results show that the entropy-only intrinsic reward performs best followed by an uncertainty-based intrinsic reward. We hypothesize that the reason why a simple entropy-only intrinsic reward works well is that state-skill CPC representation learning clusters similar behaviors together. Since similar behaviors are clustered, maximizing the entropy of state-transition embeddings produces increasingly diverse behaviors.

|        | disc.           | similarity      | uncertainty     | entropy         | APS             |
|--------|-----------------|-----------------|-----------------|-----------------|-----------------|
| walker | $0.79 \pm 0.04$ | $0.79 \pm 0.03$ | $0.76 \pm 0.04$ | $0.82 \pm 0.02$ | $0.50 \pm .04$  |
| quad.  | $0.45 \pm 0.07$ | $0.60 \pm 0.05$ | $0.70 \pm 0.03$ | $0.76 \pm 0.03$ | $.48 \pm 0.02$  |
| mean   | 0.62            | 0.70            | 0.73            | 0.80            | 0.49            |

Table 1: Comparing different potential intrinsic rewards for CIC, we find that entropy-based intrinsic reward performs best, suggesting that the CIC discriminator is primarily useful for representation learning. These are normalized scores averaged over 5 seeds across 8 downstream tasks. Note that all intrinsic reward specifications outperform the baseline methods. Since the particle entropy estimates a quantity proportional to the entropy, two-term intrinsic rewards need to be carefully balanced with a hyperparameter. We believe this is the reason the various intrinsic rewards perform worse than entropy-only one.

**The importance of representation learning:** To what extent does representation learning with the state-skill CIC loss affect the agent's exploration capability? To answer this question we train the CIC agent with the entropy intrinsic reward with and without the representation learning auxiliary loss for 2M steps. The zero-shot reward plotted in Fig. 6 indicates that without representation learning the policy collapses. With representation learning, the agent is able to discover diverse skills evidenced by the non-zero reward. This result suggests that state-skill CPC representation learning is a critical part of CIC.

**Qualitative analysis of CIC behaviors:** Qualitatively, we find that CIC is able to learn locomotion behaviors in DMC without extrinsic information such as early termination as in OpenAI Gym. While most skills are higher entropy and thus more chaotic, we show in Fig 1 that structured behaviors can be isolated by fixing a particular skill vector. For example, in the walker and quadruped domains - balancing, walking, and flipping skills can be isolated. For more qualitative investigations we refer the reader to Appendix H.

**Skill architecture and adaptation ablations:** We find that projecting the skill to a latent space before inputting it as the key for the contrastive loss is an important design decision (see Fig. 5a), most likely because this reduces the diversity of the skill vector making the discriminator task simpler.

We also find empirically that the skill dimension is an important hyperparameter and that larger skills results in better zero-shot performance (see Fig. 5b), which empirically supports the hypothesis posed in Section 2 and Appendix G that larger skill spaces are important for internalizing diverse behaviors. Interestingly, CIC zero-shot performance is poor in lower skill dimensions (e.g. $\dim(z) < 10$),

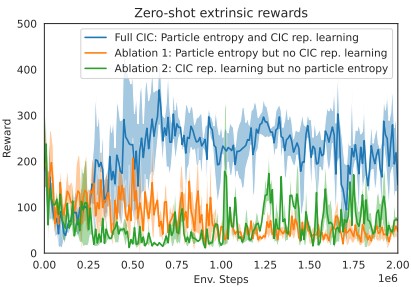

Figure 6: Mean zero-shot extrinsic rewards for Quadruped stand over 3 seeds with and without state-skill representation learning. Without representation learning, the algorithm collapses. Similarly, with CIC representation learning but no entropy term (in which case we use the discriminator as the intrinsic reward) the policy also collapses. Note that there is no finetuning happening here. We're showing the task-specific extrinsic reward during reward-free pre-training as a way to sense-check exploration policy.

suggesting that when $\dim(z)$ is small CIC performs no better than prior competence-based methods such as DIAYN, and that scaling to larger skills enables CIC to pre-train effectively.

To measure the effect of skill finetuning described in Section 4, we sweep mean skill values along the interval of the uniform prior $[0, 1]$ with a budget of 4k total environment interactions and read out the performance on the downstream task. By sweeping, we mean simply iterating over the interval $[0, 1]$ with fixed step size (e.g. $v = 0, 0.1, \ldots, 0.9, 1$) and setting $z_i = v$ for all $i$. This is not an optimal skill sampling strategy but works well due to the extremely limited number of samples for skill selection. We evaluate this ablation on the Quadruped Stand and Run downstream tasks. The results shown in Fig. 5 indicate that skill selection can substantially affect zero-shot downstream task performance.

## 7 Related Work

The most closely related prior algorithms to CIC are APT [15] and APS [27]. Both CIC and APS use the $\mathcal{H}(\tau) - \mathcal{H}(\tau|z)$ decomposition of the mutual information and both used a particle estimator [22] to compute the state entropy as in [15]. The main difference between CIC and APS is the discriminator. APS uses successor features as in [31] for its discriminator while CIC uses a noise contrastive estimator. Unlike successor features, which empirically only accommodate low-dimensional continuous skill spaces (see Table 2), the noise contrastive discriminator is able to leverage higher continuous dimensional skill vectors. Like APT, CIC has an intrinsic reward that maximizes $\mathcal{H}(\tau)$. However, CIC also does contrastive skill learning to shape the embedding space and outputs a skill-conditioned policy.

The CIC discriminator is similar to the one used in DISCERN [36], a goal-conditioned unsupervised RL algorithm. Both methods use a contrastive discriminator by sampling negatives and computing an inner product between queries and keys. The main differences are (i) that DISCERN maximizes $I(\tau; g)$ where $g$ are image goal embeddings while CIC maximizes $I(\tau; z)$ where $z$ are abstract skill vectors; (ii) DISCERN uses the DIAYN-style decomposition $I(\tau; g) = H(g) - H(g|\tau)$ while CIC decomposes through $H(\tau) - H(\tau|z)$, and (iii) DISCERN discards the $H(g)$ term by sampling goals uniformly while CIC explicitly maximizes $\mathcal{H}(\tau)$. While DISCERN and CIC share similarities, DISCERN operates over image goals while CIC operates over abstract skill vectors so the two methods are not directly comparable.

Finally, another similar algorithm to CIC is DADS [20] which also decomposes through $H(\tau) - H(\tau|z)$. While CIC uses a contrastive density estimate for the discriminator, DADS uses a maximum likelihood estimator similar to DIAYN. DADS maximizes $I(s'|s, z)$ and estimates entropy $\mathcal{H}(s'|s)$ by marginalizing over $z$ such that $\mathcal{H}(s'|s) = -\log \sum_i q(s'|s, z_i)$ while CIC uses a particle estimator.

Table 2: Competence-based Unsupervised Skill Discovery Algorithms

| Algorithm | Intrinsic Reward | Decomposition | Explicit $\max \mathcal{H}(\tau)$ | Skill Dim. | Skill Space |
|---|---|---|---|---|---|
| SSN4HRL [42] | $\log q_\psi(z\|s_t)$ | $H(z) - H(z\|\tau)$ | No | 6 | discrete one-hot |
| VIC [18] | $\log q_\psi(z\|s_H))$ | $H(z) - H(z\|\tau)$ | No | 60 | discrete one-hot |
| VALOR [25] | $\log q_\psi(z\|s_{1:H})$ | $H(z) - H(z\|\tau)$ | No | 64 | discrete one-hot |
| DIAYN [17] | $\log q_\psi(z\|s_t)$ | $H(z) - H(z\|\tau)$ | No | 128 | discrete one-hot |
| DADS [20] | $q_\psi(s'\|z,s) - \sum_i \log q(s'\|z_i,s)$ | $H(\tau) - H(\tau\|z)$ | Yes | 5 | continuous |
| VISR [31] | $\log q_\psi(z\|s_t)$ | $H(z) - H(z\|\tau)$ | No | 10 | continuous |
| APS [27] | $F_{\text{Successor}}(s\|z) + \mathcal{H}_{\text{particle}}(s)$ | $\mathcal{H}(\tau) - \mathcal{H}(\tau\|z)$ | Yes | 10 | continuous |
| CIC | $F_{\text{CIC}}(s,s'\|z) + \mathcal{H}_{\text{particle}}(s,s')$ | $\mathcal{H}(\tau) - \mathcal{H}(\tau\|z)$ | Yes | 64 | continuous |

Table 3: A list of competence-based algorithms. We describe the intrinsic reward optimized by each method and the decomposition of the mutual information utilized by the method. We also note whether the method explicitly maximizes state transition entropy. Finally, we note the maximal dimension used in each work and whether the skills are discrete or continuous. All methods prior to CIC only support small skill spaces, either because they are discrete or continuous but low-dimensional.

# 8 Limitations and Impact

While CIC achieves leading results on state-based URLB, we would also like to address its limitations. First, in this paper we only consider MDPs (and not partially observed MDPs) where the full state is observable. We focus on MDPs because generating diverse behaviors in environments with large state spaces has been the primary bottleneck for competence-based exploration. Combining CIC with visual representation learning to scale this method to pixel-based inputs is a promising future direction for research not considered in this work.

One issue with unsupervised RL algorithms (and hence CIC) in terms of potentially negative societal impact is that self-supervised exploration can be dangerous. Since self-supervised agents maximize intrinsic rewards, this can lead to destructive behavior. For example, when deploying CIC on a Walker or Quadruped robot it learns chaotic exploration behaviors [6] that would most likely break the robot in real-world settings. Alignment of exploration agents to prevent them from learning dangerous policies is a promising direction for future work.

# 9 Conclusion

We have introduced a new competence-based algorithm – Contrastive Intrinsic Control (CIC) – which enables more effective exploration than prior unsupervised skill discovery algorithms by explicitly encouraging diverse behavior while distilling predictable behaviors into skills with a contrastive discriminator. We showed that CIC is the first competence-based approach to achieve leading performance on URLB. We hope that this encourages further research in developing RL agents capable of generalization.

# 10 Acknowledgements

We would like to thank Ademi Adeniji, Xinyang Geng, Fangchen Liu for helpful discussions. We would also like to thank Phil Bachman for useful feedback. This work was partially supported by Berkeley DeepDrive, NSF AI4OPT AI Institute for Advances in Optimization under NSF 2112533, and the Office of Naval Research grant N00014-21-1-2769.

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
