# A Competence-based Exploration Algorithms

The competence-based algorithms considered in this work aim to maximize $I(\tau; s)$. The algorithms differ by ho they decompose mutual information, whether they explicitly maximize behavioral entropy, their skill space (discrete or continuous) and their intrinsic reward structure. We provide a list of common competence-based algorithms in Table 2.

# B Deep Deterministic Policy Gradient (DDPG)

A DDPG is an actor-critic RL algorithm that performs off-policy gradient updates and learns a Q function $Q_\phi(s, a)$ and an actor $\pi_\theta(a|s)$. The critic is trained by satisfying the Bellman equation.

$$\mathcal{L}_Q(\phi, \mathcal{D}) = \mathbb{E}_{(s_t, a_t, r_t, s_{t+1}) \sim \mathcal{D}} \left[ \left( Q_\phi(s_t, a_t) - r_t - \gamma Q_{\bar{\phi}}(s_{t+1}, \pi_\theta(s_{t+1})) \right)^2 \right]. \tag{6}$$

Here, $\bar{\phi}$ is the Polyak average of the parameters $\phi$. As the critic minimizes the Bellman error, the actor maximizes the action-value function.

$$\mathcal{L}_\pi(\theta, \mathcal{D}) = \mathbb{E}_{s_t \sim \mathcal{D}} \left[ Q_\phi(s_t, \pi_\theta(s_t)) \right]. \tag{7}$$

# C Baselines

For baselines, we choose the existing set of benchmarked unsupervised RL algorithms on URLB. We provide a quick summary of each method. For more detailed descriptions of each baseline we refer the reader to URLB [21]

*Competence-based Baselines:* CIC is a competence-based exploration algorithm. For baselines, we compare it to DIAYN [17], SMM [26], and APS [27]. Each of these algorithms is described in Table 2. Notably, APS is a recent state-of-the-art competence-based method that is the most closely related algorithm to the CIC algorithm.

*Knowledge-based Baselines:* For knowledge-based baselines, we compare to ICM [12], Disagreement [13], and RND [14]. ICM and RND train a dynamics model and random network prediction model and define the intrinsic reward to be proportional to the prediction error. Disagreement trains an ensemble of dynamics models and defines the intrinsic reward to be proportional to the uncertainty of an ensemble.

*Data-based Baselines:* For data-based baselines we compare to APT [15] and ProtoRL [16]. Both methods use a particle estimator to estimate the state visitation entropy. ProtoRL also performs discrete contrastive clustering as in [43] as an auxiliary task and uses the resulting clusters to compute the particle entropy. While ProtoRL is more effective than APT when learning from pixels, on state-based URLB APT is competitive with ProtoRL. Our method CIC is effectively a skill-conditioned APT agent with a contrastive discriminator.

# D   Full CIC Algorithm

The full CIC algorithm with both pre-training and fine-tuning phases is shown in Algorithm 1. We pre-train CIC for 2M steps, and finetune it on each task for 100k steps.

---

**Algorithm 1** Contrastive Intrinsic Control

---

**Require:** Initialize all networks: encoders $g_{\psi_1}$ and $g_{\psi_2}$, actor $\pi_\theta$, critic $Q_\phi$, replay buffer $\mathcal{D}$.
**Require:** Environment (env), $M$ downstream tasks $T_k, k \in [1, \ldots, M]$.
**Require:** pre-train $N_{\mathrm{PT}} = 2M$ and fine-tune $N_{\mathrm{FT}} = 100K$ steps.
  1: **for** $t = 1..N_{\mathrm{PT}}$ **do**                         ▷ Part 1: Unsupervised Pre-training
  2:     Sample and encode skill $z \sim p(z)$ and $z \leftarrow g_{\psi_2}(z)$
  3:     Encode state $s_t \leftarrow g_{\psi_1}(s_t)$ and sample action $a_t \leftarrow \pi_\theta(s_t, z) + \epsilon$ where $\epsilon \sim \mathcal{N}(0, \sigma^2)$
  4:     Observe next state $s_{t+1} \sim P(\cdot | s_t, a_t)$
  5:     Add transition to replay buffer $\mathcal{D} \leftarrow \mathcal{D} \cup (s_t, a_t, s_{t+1})$
  6:     Sample a minibatch from $\mathcal{D}$, compute contrastive loss in Eq.4 and update encoders $g_{\psi_1}, g_{\psi_2}$, compute
      CIC intrinsic reward with Eq. 5 and update actor $\pi_\theta$ and critic $Q_\phi$
  7: **end for**
  8: **for** $T_k \in [T_1, \ldots, T_M]$ **do**                    ▷ Part 2: Supervised Fine-tuning
  9:     Initialize all networks with weights from pre-training phase and an empty replay buffer $\mathcal{D}$.
 10:     **for** $t = 1 \ldots 4,000$ **do**
 11:         Take random action $a_t \sim \mathcal{N}(0, 1)$
 12:         Select skill with grid sweep over unit interval $[0, 1]$ every 100 steps
 13:         Sample minibatch from $\mathcal{D}$ and update actor $\pi_\theta$ and critic $Q_\phi$
 14:     **end for**
 15:     Fix skill $z$ that achieved highest extrinsic reward during grid sweep.
 16:     **for** $t = 4,000 \ldots N_{\mathrm{FT}}$ **do**
 17:         Encode state $s_t \leftarrow g_{\psi_1}(s_t)$ and sample action $a_t \leftarrow \pi_\theta(s_t, z) + \epsilon$ where $\epsilon \sim \mathcal{N}(0, \sigma^2)$
 18:         Observe next state and reward $s_{t+1}, r_t^{\mathrm{ext}} \sim P(\cdot | s_t, a_t)$
 19:         Add transition to replay buffer $\mathcal{D} \leftarrow \mathcal{D} \cup (s_t, a_t, r_t^{\mathrm{ext}}, s_{t+1})$
 20:         Sample minibatch from $\mathcal{D}$ and update actor $\pi_\theta$ and critic $Q_\phi$.
 21:     **end for**
 22:     Evaluate performance of RL agent on task $T_k$
 23: **end for**

---

# E  Hyper-parameters

Baseline hyperparameters are taken from URLB [21], which were selected by performing a grid sweep over tasks and picking the best performing set of hyperparameters. Except for the skill dimension, hyperparameters for CIC are borrowed from URLB.

Table 4: Hyper-parameters used for CIC .

| DDPG hyper-parameter | Value |
|---|---|
| Replay buffer capacity | $10^6$ |
| Action repeat | 1 |
| Seed frames | 4000 |
| $n$-step returns | 3 |
| Mini-batch size | 1024 |
| Seed frames | 4000 |
| Discount ($\gamma$) | 0.99 |
| Optimizer | Adam |
| Learning rate | $10^{-4}$ |
| Agent update frequency | 2 |
| Critic target EMA rate ($\tau_Q$) | 0.01 |
| Features dim. | 1024 |
| Hidden dim. | 1024 |
| Exploration stddev clip | 0.3 |
| Exploration stddev value | 0.2 |
| Number pre-training frames | $2 \times 10^6$ |
| Number fine-turning frames | $1 \times 10^5$ |
| CIC hyper-parameter | Value |
| Skill dim | 64 continuous |
| Prior | Uniform [0,1] |
| Skill sampling frequency (steps) | 50 |
| State net arch. $g_{\psi_1}(s)$ | $\dim(\mathcal{O}) \to 1024 \to 1024 \to 64$ ReLU MLP |
| Skill net arch. $g_{\psi_2}(z)$ | $64 \to 1024 \to 1024 \to 64$ ReLU MLP |
| Prediction net arch. | $64 \to 1024 \to 1024 \to 64$ ReLU MLP |

| Statistic | ICM | Dis. | RND | APT | Proto | DIAYN | APS | SMM | CIC | % CIC > APS | % CIC > Proto |
|---|---|---|---|---|---|---|---|---|---|---|---|
| Median ↑ | 0.45 | 0.56 | 0.58 | 0.62 | 0.66 | 0.44 | 0.47 | 0.22 | 0.76 | +61% | +15% |
| IQM ↑ | 0.41 | 0.51 | 0.61 | 0.65 | 0.65 | 0.40 | 0.43 | 0.25 | 0.77 | +79% | +18% |
| Mean ↑ | 0.43 | 0.51 | 0.63 | 0.66 | 0.65 | 0.44 | 0.46 | 0.35 | 0.76 | +65% | +17% |
| OG ↓ | 0.57 | 0.49 | 0.37 | 0.35 | 0.35 | 0.56 | 0.54 | 0.65 | 0.24 | -44% | -68% |

Table 5: Statics for downstream task normalized scores for CIC and baselines from URLB [21]. CIC improves over both the prior leading competence-based method APS [27] and overall next-best exploration algorithm ProtoRL [16] across all readout statistics. Each data point is a statistic computed using 10 seeds and 12 downstream tasks (120 experiments per data point). The statistics are computed using RLiable [35].

# F  Raw Numerical Results

We provide a list of raw numerical results for finetuning CIC and baselines in Tables 5 and 6. All baselines were run using the code provided by URLB [21] for 10 seeds per downstream task.

| | | | Pre-trainining for $2 \times 10^6$ environment steps | | | | | | | | | |
|---|---|---|---|---|---|---|---|---|---|---|---|---|
| Domain | Task | Expert | DDPG | CIC | ICM | Disagreement | RND | APT | ProtoRL | SMM | DIAYN | APS |
| Walker | Flip | 799 | 538±27 | 631 ± 34 | 417±16 | 346±13 | 474±39 | 544±14 | 456±12 | 450±24 | 319±17 | 465±20 |
| | Run | 796 | 325±25 | 486 ± 25 | 247±21 | 208±15 | 406±30 | 392±26 | 306±13 | 426±26 | 158±8 | 134±16 |
| | Stand | 984 | 899±23 | 959 ± 2 | 859±23 | 746±34 | 911±5 | 942±6 | 917±27 | 924±12 | 695±46 | 721±44 |
| | Walk | 971 | 748±47 | 885 ± 28 | 627±42 | 549±37 | 704±30 | 773±70 | 792±41 | 770±44 | 498±27 | 527±79 |
| Quadruped | Jump | 888 | 236±48 | 595 ± 42 | 178±35 | 389±62 | 637±12 | 648±18 | 617±44 | 96±7 | 660±43 | 463±51 |
| | Run | 888 | 157±31 | 505 ± 47 | 110±18 | 337±30 | 459±6 | 492±14 | 373±33 | 96±6 | 433±29 | 281±17 |
| | Stand | 920 | 392±73 | 761 ± 54 | 312±68 | 512±89 | 766±43 | 872±23 | 716±56 | 123±11 | 851±43 | 542±53 |
| | Walk | 866 | 229±57 | 723 ± 43 | 126±27 | 293±37 | 536±39 | 770±47 | 412±54 | 80±6 | 576±81 | 436±79 |
| Jaco | Reach bottom left | 193 | 72±22 | 138 ± 9 | 111±11 | 124±7 | 110±5 | 103±8 | 129±8 | 45±7 | 39±6 | 76±8 |
| | Reach bottom right | 203 | 117±18 | 145 ± 7 | 97±9 | 115±10 | 117±7 | 100±6 | 132±8 | 46±11 | 38±5 | 88±11 |
| | Reach top left | 191 | 116±22 | 153 ± 7 | 82±14 | 106±12 | 99±6 | 73±12 | 123±9 | 36±3 | 19±4 | 68±6 |
| | Reach top right | 223 | 94±18 | 163 ± 4 | 103±11 | 139±7 | 100±6 | 90±10 | 159±7 | 47±6 | 28±6 | 76±10 |

Table 6: Performance of CIC and baselines on state-based URLB after first pre-training for $2 \times 10^6$ steps and then finetuning with extrinsic rewards for $1 \times 10^5$. All baselines were run for 10 seeds per downstream task for each algorithm using the code provided by URLB [21]. A total of $1080 = 9$ algorithms $\times$ 12 tasks $\times$ 10 seeds experiments were run.

# G    Toy Example to Illustrate the Need for Larger Skill Spaces

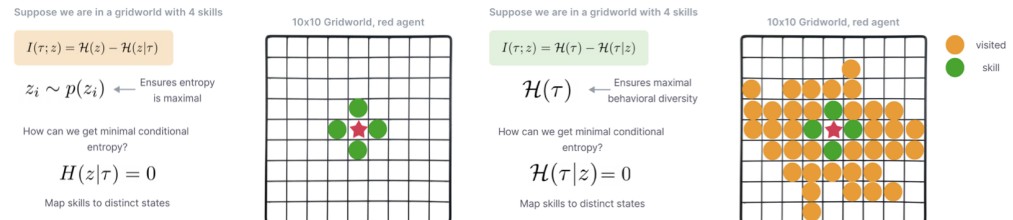

Figure 7: A gridworld example motivating the need for large skill spaces. In this environment, we place an agent in a $10 \times 10$ gridworld and provide the agent access to four discrete skills. We show that the mutual information objective can be maximized by mapping these four skills to the nearest neighboring states resulting in low behavioral diversity and exploring only four of the hundred available states.

We illustrate the need for larger skill spaces with a gridworld example. Suppose we have an agent in a $10 \times 10$ sized gridworld and that we have four discrete skills at our disposal. Now let $\tau = s$ and consider how we may achieve maximal $I(\tau; z)$ in this setting. If we decompose $I(\tau; z) = \mathcal{H}(z) - \mathcal{H}(z|\tau)$ then we can achieve maximal $\mathcal{H}(z)$ by sampling the four skills uniformly $z \sim p(z)$. We can achieve $\mathcal{H}(z|\tau) = 0$ by mapping each skill to a distinct neighboring state of the agent. Thus, our mutual information is maximized but as a result the agent only explores four out of the hundrend available states in the gridworld.

Now suppose we consider the second decomposition $I(\tau; z) = \mathcal{H}(\tau) - \mathcal{H}(\tau|z)$. Since the agent is maximizing $\mathcal{H}(\tau)$ it is likely to visit a diverse set of states at first. However, as soon as it learns an accurate discriminator we will have $\mathcal{H}(\tau|z)$ and again the skills can be mapped to neighboring states to achieve minimal conditional entropy. As a result, the skill conditioned policy will only be able to reach four out of the hundrend possible states in this gridworld. This argument is shown visually in Fig. 7.

Skill spaces that are too large can also be an issue. Consider if we had $100$ skills at our disposal in the same gridworld. Then the agent could minimize the conditional entropy by mapping each skill to a unique state which would result in the agent memorizing the environment by finding a one-to-one mapping between states and skills. While this is a potential issue it has not been encountered in practice yet since current competence-based methods support small skill spaces relative to the observation space of the environment.

# H    Qualitative Analysis of Skills

We provide two additional qualitative analyses of behaviors learned with the CIC algorithm. First, we take a simple pointmass setting and set the skill dimension to 1 in order to ablate the skills learned by the CIC agent in a simple setting. We sweep over different values of $z$ and plot the behavioral flow vector field (direction in which point mass moves) in Fig.8. We find that the pointmass learns skills that produce continuous motion and that the direction of the motion changes as a function of the skill value. Near the origin the pointmass learns skills that span all directions, while near the edges the point mass learns to avoid wall collisions. Qualitatively, many behaviors are periodic.

Qualitatively, we find that methods like DIAYN that only support low dimensional skill vectors and do not explicitly incentivize diverse behaviors in their objective produce policies that map skills to a small set of static behaviors. These behaviors shown in Fig. 9 are non-trivial but also have low behavioral diversity and are not particularly useful for solving the downstream task. This observation is consistent with [44] where the authors found that DIAYN maps to static "yoga" poses in DeepMind Control. In contrast, behaviors produce by CIC are dynamic resulting flipping, jumping, and locomotive behaviors that can then be adapted to efficiently solve downstream tasks.

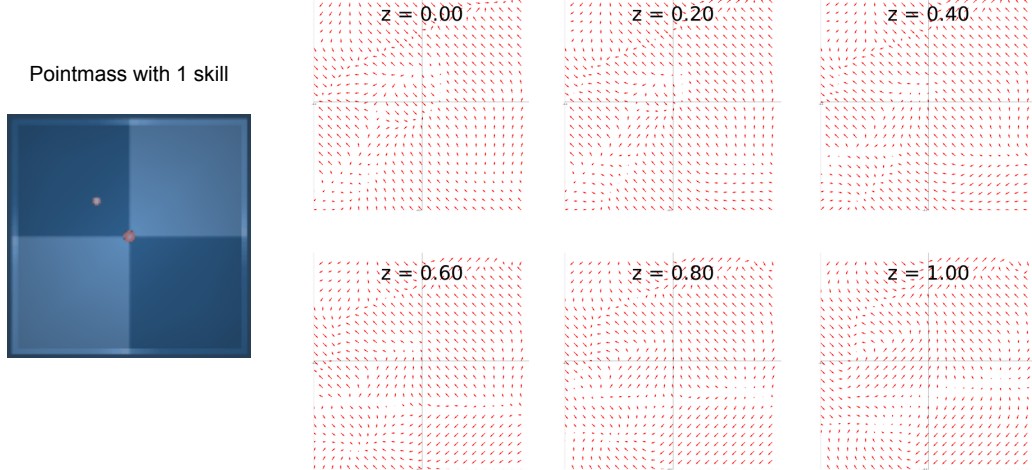

Figure 8: Learning curves for finetuning pre-trained agents for 100k steps. Task performance is aggregated for each domain, such that each curve represents the mean normalized scores over $4 \times 10 = 40$ seeds. The shaded regions represent the standard error. CIC surpasses the performance of the prior state-of-the-art on Walker and Jaco tasks while tying on Quadruped. CIC is the only algorithm that performs consistently well across all three domains.

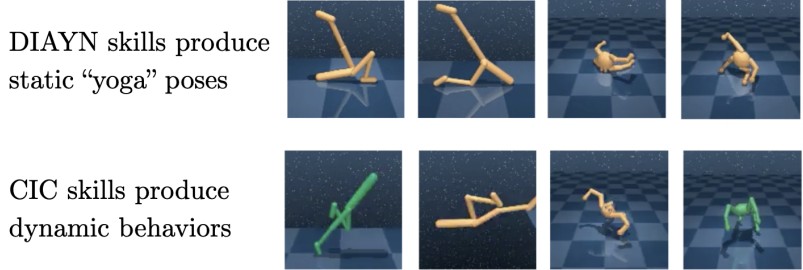

Figure 9: Qualitative visualization of DIAYN and CIC pre-training on the Walker and Quadruped domains from URLB. Confirming findings in prior work [44], we also find that DIAYN policies produce static but non-trivial behaviors mapping to "yoga" poses while CIC produces diverse and dynamic behaviors such as walking, flipping, and standing. Though it's hard to see from these images, all the DIAYN skills get stuck in frozen poses while the CIC skills are producing dynamic behavior with constant motion.

# I  OpenAI Gym vs. DeepMind control: How Early Termination Leaks Extrinsic Signal

Prior work on unsupervised skill discovery for continuous control [17, 20] was evaluated on OpenAI Gym [32] and showed diverse exploration on Gym environments. However, Gym environment episodes terminate early when the agent loses balance, thereby leaking information about the extrinsic task (e.g. balancing or moving). However, DeepMind Control (DMC) episodes have a fixed length of 1k steps. In DMC, exploration is therefore harder since the agent needs to learn to balance without any extrinsic signal.

To evaluate whether the difference in the two environments has impact on competence-based exploration, we run DIAYN on the hopper environments from both Gym and DMC. We compare to ICM, a popular exploration baseline, and a Fixed baseline where the agent receives an intrinsic reward of 1 for each timestep and no algorithms receive extrinsic rewards. We then measure the extrinsic reward, which loosely corresponds to the diversity of behaviors learned. Our results in Fig. 3 show that indeed DIAYN is able to learn diverse behaviors in Gym but not in DMC while ICM is able to learn diverse behaviors in both environments. Interestingly, the Fixed baseline achieves the highest

reward on the Gym environment by learning to stand and balance. These results further motivate us to evaluate on URLB which is built on top of DMC.

## J   CIC vs Other Types of Contrastive Learning for RL

Contrastive learning in CIC is different than prior vision-based contrastive learning in RL such as CURL [45], since we are not performing contrastive learning over augmented images but rather over state transitions and skills. The contrastive objective in CIC is used for unsupervised learning of behaviors while in CURL it is used for unsupervised learning of visual features.

We provide pseudocode for the CIC loss below:

```
def discriminator_loss(states, next_states, skills, temp):
    """
    - states and skills are sampled from replay buffer
    - skills were sampled from uniform dist [0,1] during agent rollout
    - states / next_states: dim (B, D_state)
    - skills: dim (B, D_skill)
    """

    transitions = concat(states, next_states, dim=1)

    query = skill_net(skills) # (B, D_hidden) -> (B, D_hidden)
    key = transition_net(transitions) # (B, 2*D_state) -> (B, D_hidden)

    query = normalize(query, dim=1)
    key = normalize(key, dim=1)

    logits = matmul(query, key.T) / temp # (B, B)
    labels = arange(logits.shape[0])

    # positives are on diagonal, negatives are off diagonal
    # for each skill, negatives are sampled from transitions
    # while skills are fixed
    loss = cross_entropy(logits, labels)

    return loss
```

Listing 2: CIC discriminator loss

This is substantially different from prior contrastive learning works in RL such as CURL [45], which perform contrastive learning over images.

```
def curl_loss(obs, W, temp):
    """
    - observation images are sampled from replay buffer
    - obs: dim (B, C, H, W)
    - W: projection matrix (D_hidden, D_hidden)
    """

    query = aug(obs)
    key = aug(obs)

    query = cnn_net(query) # (B, D_hidden)
    key = cnn_net(key) # (B, D_hidden)

    logits = matmul(matmul(query, W), key.T) / temp # (B, B)
    labels = arange(logits.shape[0])

    # positives are on diagonal
    # negatives are off diagonal
    loss = cross_entropy(logits, labels)
```

```
21        return loss
```
Listing 3: CURL contrastive loss

## K   On estimates of Mutual Information

In this work we have presented CIC - a new competence-based algorithm that achieves leading performance on URLB compared to prior unsupervised RL methods.

One might wonder whether estimating the exact mutual information (MI) or maximizing the tightest lower bound thereof is really the goal for unsupervised RL. In unsupervised representation learning, state-of-the-art methods like CPC and SimCLR maximize the lower bound of MI based on Noise Contrastive Estimation (NCE). However, as proven in CPC [33] and illustrated in [46] NCE is upper bounded by $\log N$, meaning that the bound is loose when the MI is larger than $\log N$. Nevertheless, these methods have been repeatedly shown to excel in practice. In [47] the authors show that the effectiveness of NCE results from the inductive bias in both the choice of feature extractor architectures and the parameterization of the employed MI estimators.

We have a similar belief for unsupervised RL - that with the right parameterization and inductive bias, the MI objective will facilitate behavior learning in unsupervised RL. This is why CIC lower bounds MI with (i) the particle based entropy estimator to ensure explicit exploration and (ii) a contrastive conditional entropy estimator to leverage the power of contrastive learning to discriminate skills. As demonstrated in our experiments, CIC outperforms prior methods, showing the effectiveness of optimizing an intrinsic reward with the CIC MI estimator.

## L   Compute Resources

CIC runs on a single GPU. In our experiments we used 4 NVIDIA TITAN RTX GPUs. Pre-training one seed of CIC for 2M steps takes 12-24 hours while fine-tuning to downstream tasks for 100k steps takes 30min - 1 hour.