# OpenReview forum: "Unsupervised Reinforcement Learning with Contrastive Intrinsic Control"
_NeurIPS.cc/2022/Conference — NeurIPS 2022 Accept_

### Official Review · Reviewer_tFmb · 2022-07-07

**Rating:** 6
**Confidence:** 4
**Soundness:** 3 good
**Presentation:** 3 good
**Contribution:** 3 good

**Summary:**

This work proposes CIC, a competence-based unsupervised RL method that maximizes the mutual information between latent skills and state transitions. CIC outperforms related baselines on the Unsupervised RL benchmark (URLB) where the agent only has access to the extrinsic reward for a short adaptation phase in the downstream tasks.

**Questions:**

 1. I'm curious how the estimation of decomposed mutual information in Table 3 would distribute across different methods. And how CIC may outperform baselines.

2. The decomposition of mutual information used by CIC encourages diverse behaviors through the entropy term $H(\tau)$. I'm curious whether there is a trade-off for losing some diversity of the skills $H(z)$?

3. I see that previous works employ too small skill dimensions and it's great that CIC employs a larger skill dimension. But the skill dimension should not be too large as well, so there should be some sweet spot that may require hyperparameter tuning. Since the tasks targeted are unsupervised adaptation, does it make sense to tune the dimension without the knowledge of the diversity of the downstream tasks?

**Limitations:**

The authors have adequately addressed the limitations.

**Strengths And Weaknesses:**

***Strengths***

This study has significant strengths in the well-established experimental section.

1. Decent experimental setup

The URLB is a well-established benchmark to evaluate the unsupervised adaptation performance of RL methods. I appreciate the comparison between Gym and DMControl.

2. High statistical significance

I appreciate the authors running experiments across as many as 10 seeds, and reporting IQM scores. This provides high statistical significance for the results compared to other methods that report only mean results with 3~5 seeds. Also they compared many relevant prior works.

3. Ablations

Because there are many overlapping contributions with existing studies (APS, APT, CURL,..etc.), it is easy to give a mixed feeling of contribution. But the authors managed to prevent this through ablations in Figure 5.


***Weaknesses***

The weaknesses of this work lie in the originality and motivation. As this is a completely experimental paper, I understand that theoretically, it is difficult to prove the strength of this study. I felt the connection between the motivation and contribution seemed a bit vague.

1. Originality

As the authors are aware, there were previous works that use particle-based entropy or contrastive learning for unsupervised learning (APT, APS, CURL). The novelty of CIC is that it is the first that used contrastive learning between state transition and latent skill. The absence of a related work section may make it feel that novelty is lacking. From this point of view, I appreciate the comparison made in Appendix D. Maybe it would be better if the authors add such content to the main manuscript with a little more detail.

2. Motivation

The two points made in the motivation section ("Competence-based algorithms do not ensure diverse behaviors", "Why it is important to utilize high-dimensional skills") are indeed very important and useful points. But there seems to be a lack of a link between these motivations and the contrastive learning and particle-based entropy. Some questions remain about whether contrastive learning and particle-based entropy are the "only" or the "best" ways to estimate $H(\tau|z)$ and $H(\tau)$, respectively. Some of these concerns seem to be addressed in Appendix L, which I think would be better to be in the main manuscript.

---

> ### Author Response · Authors · 2022-08-02
> **Thank you for reviewing our submission, we address your questions below**
>
> Thank you very much for reviewing our work. We’re glad the reviewer found that the paper has significant strengths in experimental evaluation, high statistical significance, and insightful ablations.
>
> We address the reviewers points of feedback and questions below:
>
> **The absence of a related work section may make it feel that novelty is lacking.**
>
> Thank you for pointing this out. We have moved the related work section from the appendix to the main paper to better contextualize our work.
>
> **There seems to be a lack of a link between these motivations and the contrastive learning and particle-based entropy.**
>
> Indeed, as the reviewer points out the two motivations stated are (1) prior competence-based methods do not ensure diversity and (2) why it is important to learn high-dimensional skills. The mapping between these and the contrastive learning / particle-based entropy is:
>
> *High-dimensional skills <-> contrastive learning:* State-skill contrastive learning is what enables our method to accommodate high-dimensional skill where others could not (see main results as well as ablation in Fig 5b).
>
> *Behavior diversity <-> particle entropy estimator:* Prior works such as DIAYN ensure skill diversity but not behavior diversity, where by behavior diversity we mean the number of unique states discovered with the agent's policy. The reason is that large $H(z)$ does not mean necessarily that $H(\tau)$ is large. CIC (like APS) uses the particle entropy estimator to ensure maximal $H(\tau)$ by maximizing this quantity directly. However, APS only supports small dimensional skills which limits its performance.
>
> We argue both points are required and show this explicitly in Figure 6.
>
> ### Questions
>
> **Q1:** *I'm curious how the estimation of decomposed mutual information in Table 3 would distribute across different methods. And how CIC may outperform baselines.*
>
> This is an interesting question regarding how the various methods would benefit from the alternate decomposition of MI from the ones they use. However, it is not clear if such a comparison is possible to make. The reason is that methods that choose the $I(\tau;z) = H(z) - H(z|\tau)$ decomposition explicitly sample $z$ from some noise distribution (e.g. uniform) to ensure maximal entropy. However, there is no way to sample $\tau$ from noise since $\tau$ is given by the environment after interaction.
>
> For example, to go from $I(\tau;z) = H(z) - H(z|\tau)$ to $I(\tau;z) = H(\tau) - H(\tau|z)$ you would need to figure out how to estimate $H(\tau)$ and maximize it, which would alter the underlying method.
>
> **Q2:** *The decomposition of mutual information used by CIC encourages diverse behaviors through the entropy term $H(\tau)$. I'm curious whether there is a trade-off for losing some diversity of the skills $H(z)$?*
>
> Indeed, there is likely a tradeoff where $H(z)$ is not strictly maximal since it is not being explicitly maximized. However, since the contrastive learning loss brings embeddings of $\tau$ and $z$ closer together, it is possible that by maximizing $H(\tau)$ you are also indirectly maximizing $H(z)$. We agree that understanding the trade-off further is an interesting line of inquiry for future work.
>
> **Q3:** *I see that previous works employ too small skill dimensions and it's great that CIC employs a larger skill dimension. But the skill dimension should not be too large as well, so there should be some sweet spot that may require hyperparameter tuning. Since the tasks targeted are unsupervised adaptation, does it make sense to tune the dimension without the knowledge of the diversity of the downstream tasks?*
>
> Yes this is a good point. Indeed, we saw in Figure 5b that the skill dimension gets saturated at some point and performance starts degrading after a 64 dim skill vector. We believe this is more of a property of the environment / task complexity, as you suggest. DeepMind control is a relatively small environment, and much larger environments may need much larger skill dimensions to capture diverse behaviors.

---

> > ### Comment · Reviewer_tFmb · 2022-08-06
> > **Thank you for the response**
> >
> > I appreciate the authors for the kind response. Although the trade-off between the diversity of the behaviors and skills is left as a future work, most of my concerns are well addressed.

---

### Official Review · Reviewer_yva3 · 2022-07-12

**Rating:** 7
**Confidence:** 4
**Soundness:** 3 good
**Presentation:** 2 fair
**Contribution:** 3 good

**Summary:**

This paper presents a contrastive-learning based method that encourages behavioural diversity for unsupervised RL (where unsupervised means without reward). Their method operates by maximizing the mutual information between state-transitions and skill vectors. The authors provide experimental evidence with the Unsupervised Reinforcement Learning Benchmark (URLB) for the efficiency of their proposed method by examining whether pre-training agents with their method can better adapt to downstream tasks.

**Questions:**

1. In equation (2), what is the expectation over?
1. Is the second to last line of the pseudocode on page 5 correct? It's not clear that the `cross_entropy` call gives you $F_{CIC}$. In particular, it seems like it's missing the first term?
1.  In line 277 it says "most likely because this reduces the diversity of the skill vector" to justify the projection, but it's not clear why this projection is necessarily reducing the diversity of the skill vector?
1. In the "skill architecture and adaptation ablations", did you try running DIAYN with larger skills?


**Limitations:**

Both limitations and potential negative societal impacts were discussed adequately (in my opinion) in section 7.

**Strengths And Weaknesses:**

# Originality
This work is mostly an extension of the work of Liu & Abbeel 2021 ([15] and [27] in paper) where the main contribution (as acknowledged by the authors) is in a novel _discriminator_ (where the discrimination is between latent skills). Their proposed improvement is also based on existing work (e.g. the noise-contrastive estimator of Gutmann & Hyvärinen 2010 ([23] in paper), the parameterization of the function for density estimation of Chen et al. 2020 ([34] in paper)).

However, in my opinion this is an original and novel combination of existing methods towards addressing a problem for which (most of) those methods were not originally designed.

# Quality
Overall I feel the quality of the paper is reasonably good. There are a few issues with clarity that I mention below. I appreciated the ablations and sensitivity analyses done in Figure 5, as it really helped motivate the design choices made in the paper. Finally, there are a few questions related to experiment quality in the Questions section belo.

# Clarity
This is perhaps the weakest point of the paper (although Figure 2 is really nice!). The writing oscillates between proper mathematical notation and a github README file (e.g. "$r,s'\sim$ `env.step(a)`" in line 57). Further, there are some improperly specified mathematical objects in the writing, which I list below:
*  In line 53, you say "$r$" is "sampled" from `env.step(a)`, but a few lines above you stated that $r$ was the reward function.
*  In line 55 it says "$\tau(s)$ ... refers to any function of the states $s$". This is ill-defined and should be more specific. $\tau$ is a function of what to what? In line 57 you're using $\tau$ as a tuple (e.g. $\tau=(s, s')$), so it's not at all clear what it's meant to represent.
*  In line 60,  $\mathcal{Z}$ has not been introduced.
* The paragraph in lines 86-95 is not at all clear because it's not clear what $\tau$ is.
* In footnote 4 in page 4 it says "Note that $\tau$ is not a trajectory but some function of states" which, again, is not at all clear what it's meant to represent.
* It's not clear what is meant in lines 173-174. In particular what are "negatives" and "positives"? Is $N$ the same thing as $N_k$ in the equation?
*  One method of finetuning is presented in lines 195-204, but another one is presented in lines 223-224. It's not clear which one was actually used for the experiments, and if both were used, when each was used.
* The histogram in Figure 5(d) (and accompanying discussion) is not clear. You're evaluating over a pre-selected grid of latents, and the histogram shows the performance of each of these grids? It's not clear how this is measuring the effect of skill fine-tuning?

# Significance
This work is addressing an important problem for a large portion of RL research (lifelong learning, zero-shot transfer, generalizability, etc.), and as such can have an important impact on the community. The empirical evaluations presented in Figure 5 are quite useful, as they help shed light on the most necessary components of the proposed algorithm.

---

> ### Author Response · Authors · 2022-08-02
> **Thank you very much for reviewing our work, we address your questions below**
>
> Thank you for reviewing our work. We very much appreciate your feedback! We’re glad the reviewer found this paper novel, the ablations and sensitivity analyses insightful, and the overall problem the paper addresses important.
>
> We address questions posed by the reviewer below. We have also edited the manuscript to incorporate each point of clarification raised.
>
> ### Points of clarification (incorporated into updated manuscript)
>
> **Line 53 - reward sampled from environment notation.**
>
> You’re right, this is unclear. The agent computes its own intrinsic reward after taking a step in the environment.
>
> **Lines 55 and 86-95- confusing definition $\tau$**
>
> Here we are just introducing a convenience variable. We apologize we made this more confusing than it needed to be, $\tau = (s,s’)$ is just a transition of two consecutive states. We have edited this line to make it clearer.
>
> We hope lines 86-95 are clearer now that $\tau$ is better defined.
>
> **In line 60, $z \in \mathcal{Z}$**
>
> This is the skill set, which can be a discrete or continuous real-valued vector space, similar to how actions belong to the action set $a \in \mathcal {A}$.
>
> **Lines 173-174 - positives and negatives**
>
> We apologize - the notation of $N$ was supposed to be under the $F_\text{CIC}$ equation (now edited). $N$ and $N_k$ are two different quantities - $N_k$ is the number of k-nearest neighbors in the particle estimator (eq 4) while $N-1$ is the number of negatives in the contrastive loss ($F_\text{CIC}$).
>
> **The histogram in Figure 5(d)**
>
> Here we simply set all values of the latent vector $z$ to the number specified on the x-axis and perform a sweep. This plot is showing that different skill values produce behaviors which result in different (zero-shot) extrinsic rewards.
>
> ### Questions
>
> Q1: In equation (2), what is the expectation over?
>
> The expectation is over the joint distribution over $z$ and $\tau$. We edited the equation to make it clearer. This variational lower bound is derived in [Barber and Agakov](https://proceedings.neurips.cc/paper/2003/file/a6ea8471c120fe8cc35a2954c9b9c595-Paper.pdf) (see eq. 3 in the referenced paper).
>
> Q2: Is the second to last line of the pseudocode on page 5 correct? It's not clear that the cross_entropy call gives you F_CIC. In particular, it seems like it's missing the first term?
>
> Yes, it is correct. The softmax cross entropy is the log of the softmax function
>
> $$\text{softmax} (x_i) = \exp(x_i) / \sum_j \exp(x_j)$$
>
> $$ \text{cross-entropy} (x_i) = \log softmax (x_i) = x_i - \log \sum_j \exp( x_j)$$
>
> where $x_j$ are the logits. The two terms in $F_{CIC}$ correspond directly to the softmax cross entropy with the cosine similarity being used to determine the logits. Note that the $1/N$ constant in the second term disappears when you take the gradient since $\nabla_\theta \log N = 0$, so the loss does not need to include this term.
>
> Q3: In line 277 it says "most likely because this reduces the diversity of the skill vector" to justify the projection, but it's not clear why this projection is necessarily reducing the diversity of the skill vector?
>
> The original skill vector is sampled from uniform noise, so it is maximally diverse. The weights of the neural network that projects the original skill vector into a latent space get updated via backprop from the $F_{CIC}$ objective. It is likely that this projection clusters some of the input noise to make its prediction task easier (similar to how GANs embed maximally entropic noise into more structured latent spaces via the GAN loss). In CIC, the latent space is likely to be more structured than the original noise (maximally entropic) distribution.
>
> Q4: In the "skill architecture and adaptation ablations", did you try running DIAYN with larger skills?
>
> When implementing the DIAYN baseline we performed a sweep over hyperparameters, including skill dimension, and picked the skill dimension that performed best.
>
> ### In conclusion
>
> Thank you again for bring up the above points of clarification. We edited the manuscript to address these.

---

> > ### Comment · Reviewer_yva3 · 2022-08-04
> > **Thanks**
> >
> > Thank you for responding to my questions!

---

### Official Review · Reviewer_4Ffu · 2022-07-12

**Rating:** 3
**Confidence:** 4
**Soundness:** 2 fair
**Presentation:** 3 good
**Contribution:** 2 fair

**Summary:**

The paper considers the problem of exploration and skill discovery in unsupervised reinforcement learning. To this end, the paper proposes CIC, an algorithm to explore novel states as well as distill the experience into reusable skills, which can be efficiently adapted to downstream tasks. This work builds upon the ideas of several previous works that maximize the mutual information between (some function of) the visited states and the latent skill vectors. These works differ in the different design choices for estimating the mutual information (generally it is a lower bound), as succinctly summarized in the paper in lines 74-76.

CIC estimates the entropy term, $\mathcal{H}(\tau)$, using a particle estimate similar to some previous work. The contribution of the paper is the use of a contrastive density estimator, specifically the NCE loss proposed in CPC (Oord et al, 2018), for the variational term in the MI lower bound. The unsupervised pre-training phase involves optimizing the CPC loss for contrastive representation learning between state transitions and skills as well as maximizing the intrinsic reward. The fine-tuning phase involves training the agent (with a few steps) using only the external task reward.

This approach is shown to outperform similar skill discovery methods on the URLB benchmark using DDPG as the backbone, and extensive experiments investigating some design and hyperparameter choices have been performed.

**Questions:**

**The use of CPC loss as variational distribution:** The noise contrastive density estimator used for $q(\tau|z)$ and the representation learning loss is exactly the same, which is the CPC loss from (Oord et al., 2018). The CPC loss is a lower bound on the MI, so it is unclear why this should be a good choice for the variational distribution. Moreover, replacing Eq. (3) in Eq. (2) contradicts the result of (Oord et al., 2018). Since CIC frames itself within the MI maximization family of skill discovery algorithms, it is crucial that the connection between the proposed method and MI maximization is presented as clearly as possible.

**$F_{CIC}$ definition vs. implementation:** The notation in the definition of $F_{CIC}$ is not clear, and there seem to be inconsistencies between this definition and the pseudocode provided for calculating this loss. The index $i$ is not specified, but assuming it is the batch index, it would seem that $F_{CIC}$ is calculated for some $z_i$ with $\tau_i$ as the positive sample and the remaining $\tau_j$ in the batch as negative samples. The pseudocode, however, samples a batch of $\tau_i$ and a batch of $z_i$, similar to how most contrastive learning methods (simCLR, CURL etc.) implement this loss.

**Intrinsic reward specified not specified clearly:** The intrinsic reward is specified in the middle of the experimental results section as a written description, rather than explicit mathematical definitions.  Also, it is not clear which intrinsic reward is used when the authors compare CIC with the baselines or perform ablations (such as in Fig 4, 5, 6). It is appreciable that the paper considers a variety of intrinsic rewards to understand which one works best, however, I believe that specifying them clearly would help improve the readability of the paper.

**CIC used in experiments does not use intrinsic reward to maximize MI:** The paper presents CIC as part of a family of methods that maximize the MI using intrinsic reward (Table 2), however their best performing model (and presumably most of their empirical results) considers only the entropy term as the intrinsic reward. This disconnect between the theory (which describes a novel method of estimating the variational lower bound on MI), and the experiments (which seem to not use this variational lower bound) significantly hurts the quality of the paper. If my understanding is correct, the intrinsic reward closest to the MI maximization objective would be (i) discriminator (from line 253-254), which is also the worst-performing variant of CIC. In addition, the contribution of using a contrastive density estimator in the variational lower bound on MI is made redundant.

**Minor comments:**

- Typo in Line 67, should direct to Table 2 in Appendix A.
- CPC loss, referred in line 165 and numerous other places, is not defined. I assume it refers to $F_{CIC}$, which is same as the CPC loss.
- $N$ is not defined for Eq. (3), assumed to be the batch size.
- The experiment plots, Fig 4, 5, 6, are present on different pages and going back and forth between Section 6 and the plots hurts the readability of the paper. I suggest moving the plots closer to the relevant text.

**Limitations:**

The authors acknowledge some limitations of their work, particularly that the method is described for MDPs (not POMDPs), and that they do not consider visual inputs in their experiments. The authors have also given though to the potential negative societal impacts of their work (really unsupervised RL in general), which is appreciable.

The main limitations of the work as per my review are the somewhat limited novelty in the proposed method, some questions on the theoretical soundness, and the discrepancy between the method described in the theory section and the one used in the experiments. There are also some concerns regarding the notation and writing, which are relatively minor and easily fixable.

**Strengths And Weaknesses:**

The problem of exploring novel states and generalizing to multiple downstream tasks are central challenges in RL, and skill discovery algorithms provide an appealing solution to both these problems. There are several works which aim to maximize (some lower bound on) the MI between states and skill vectors. In this regard, the paper has somewhat limited novelty. This work proposes maximizing MI explicitly using CPC loss for representation learning (in addition to the intrinsic reward) and the contrastive density estimator used to estimate the conditional entropy term is also well known. Nevertheless, a combination of well-known techniques can still be valuable, provided it is supported with substantial theoretical and empirical analysis. The paper performs many experiments to showcase and analyze the proposed method, however, the theoretical justification is lacking and unclear in a few places.

The paper contextualizes their work well by providing a succinct and fairly complete overview of similar work (Table 1 is quite helpful). The authors provide intuition-based but clear motivation for their design choices in Section 2, which sets up the proposed method well. However, parts of Section 3 are not clear, particularly the use of CPC loss as the variational distribution, mismatch between the specification of $F_{CIC}$ and its computation in the pseudocode, and the connection to MI maximization. The intrinsic reward mentioned from Section 3 onwards is not defined until Section 6, which might be confusing for the reader. It is also not clear what intrinsic reward is used in the ‘default’ CIC algorithm (used in Fig 4, 5, 6) - I assume it's just the entropy term since it is shown to be the best performing variant in Section 6. If so, the discussion about the variational lower bound on MI and the contribution of using a contrastive density estimator are made redundant. The experimental results, notwithstanding the issues mentioned here, are extensive and the ablation studies for various design choices are insightful.

---

> ### Author Response · Authors · 2022-08-02
> **Thank you for reviewing CIC, we address your questions and concerns below**
>
> Thank you for reviewing our submission as well as the detailed feedback. We appreciate that the reviewer found this work addresses a central challenge in RL and that the experiments are extensive and ablations insightful.
>
> We address the reviewer’s questions and concerns below.
>
> **The CPC loss is a lower bound on the MI, so it is unclear why this should be a good choice for the variational distribution.**
>
> We motivate the choice of state-to-skill CPC loss in Section 2. A practical limitation of using the variational distribution is that the corresponding skill classification (or regression) objective does not scale well to high-dimensional skill vectors. This is supported by both prior work (see Table 2 of Appendix A). As shown in many prior works (SimCLR, CPC, MoCo, etc), the CPC loss works well with higher latent dimensions which makes it a sensible choice for state-skill representation learning, which we also observe in our empirical investigations (see Figure 5b).
>
> **The notation in the definition of  is not clear**
>
> We apologize that our notation was not clear. Indeed, the loss is implemented like SimCLR / CURL and $F_{CIC}$ reflects this but we can see how that can be a bit unclear. In $F_\text{CIC}$ - $i$ is a fixed index and $j$ represents all indices (including $i$) in the batch.
>
> You can see that the loss implements $F_\text{CIC}$ because it first generates $(B,B)$ logits where on the diagonal we have $f(\tau_i, z_i)$ while across an entire batch column we have $f(\tau_j, z_i)$ where $j$ includes all $\tau$ along the batch. The cross entropy loss is similar to the one used in most modern autograd software. It expects $(B, C)$ input where $C$ is the number of classes. In contrastive learning, $C=B$ so when it receives the $(B,B)$ logits it sums over axis=1 and takes the mean over axis=0.
>
> We have revised the manuscript to make this clearer by explicitly stating the meaning of indices $i$ and $j$ and that we are summing over the batch dimension.
>
> **Intrinsic reward specified not specified clearly**
>
> The method section describes the intrinsic reward explicitly (see Section 3 “Method” subsection titled “Intrinsic Reward”). We have also added a sentence to make it clearer.
>
> **CIC used in experiments does not use intrinsic reward to maximize MI**
>
> In this work, we first derive a representation learning objective ($F_{CIC}$) and then leave the specific choice of intrinsic reward up to empirical evaluation. In Table 1 we evaluate the most common types of intrinsic rewards in skill learning literature and find that the entropy-only reward works best which is what we use for the remainder of the work.
>
> To be clear, the mean scores in Table 1 for all intrinsic rewards considered are higher than those for prior unsupervised skill RL baselines (see Table 1 and Table 6 / we have also added the APS baseline to Table 1 to make this clearer). We believe the reason these intrinsic rewards performed worse than entropy-only was due to needing to balance two terms in the intrinsic terms with a hyperparameter instead of one, since the entropy term estimates a quantity proportional to the entropy. We have added this discussion to the text.
>
> We believe this level of empirical thoroughness around the intrinsic reward specification is helpful to the community and researchers who may want to build on this work.
>
> **Minor points**
> Thank you, we updated the manuscript with corrections.

---

> > ### Comment · Reviewer_4Ffu · 2022-08-08
> > **Thank you for the response, but some concerns remain**
> >
> > Thank you for your response. The explanations for the notation and experiments are helpful, but some outstanding issues exist which are described below.
> >
> > The primary technical contribution as claimed by the paper (lines 37-41, 119-120) is the novel estimator for the discriminator. However, there are concerns regarding the theoretical soundness and experimental results for the proposed discriminator.
> >
> > **Theoretical soundness:** My concern regarding the use of InfoNCE loss as the variational distribution seems to have been misunderstood. The paper applies InfoNCE [Oord et al., 2018], which maximizes a surrogate objective (the CPC loss) for the mutual information between two variables. The paper applies InfoNCE in two different ways:
> >
> > - $F_{CIC}$ specified in equation 4 is used for representation learning to maximize MI between $\tau$ and $z$.
> > - $\log q(\tau|z)$ in equation 3 is exactly the same as $F_{CIC}$. The InfoNCE objective itself is a lower bound on mutual information, so it seems an unusual choice for the variational distribution.
> >
> > **Experimental setting:** The performance of the discriminator method was not in question, however, devoting much of the paper towards motivating the discriminator only to have very few experimental results for the intrinsic reward that utilizes it is a major weakness of the paper.
> > Parts of Sections 1, 2, and 3 motivate the decomposition of MI, and the choice of using a noise-contrastive discriminator for the intrinsic reward in equation 2. Therefore, not utilizing this particular formulation (which is also claimed as the primary contribution) for a majority of the experimental results is troubling.
> >
> > **Intrinsic reward specification:** I reemphasize the point that the paper specifies intrinsic rewards as written descriptions, rather than explicit mathematical definitions. It is important to convey the experimental details to the reader without ambiguity.
> >
> > **Section 3 notation:** Thank you for the clarification. It seems as given in equation 4, $F_{CIC}$ should be written as a function of $(\tau_i, z_i)$ - the missing $i$ was the source of the confusion. On a related note, there seem to be some inconsistencies in notation in Section 3.1. $\tau$ is defined as a single transition tuple $(s,s')$ but then using indexed notation like $\tau_i$ suggests that it is a batch of such transitions.
> >
> > Due to the concerns highlighted above, after reading the other reviews/discussion, and re-reading the paper my rating remains unchanged.

---

> > > ### Public Comment · ~Yucheng_Yang2 · 2024-12-01
> > > **Entropy is the best, maybe only for this benchmark**
> > >
> > > Thank you for your quality review! From my perspective, the reason entropy performs the best is that their URLB benchmark is biased towards pure exploration methods (or by their definition "knowledge" or "data-based" methods). In contrast, mutual information skill learning (or using their term "competence-based" methods) excels at simple navigation tasks but struggles with the more complex downstream tasks in URLB. The current benchmark is limited and potentially biased. It would be valuable for someone to create a new benchmark that better accommodates and highlights the strengths of mutual information skill learning methods.

---

### Meta-Review · Area_Chair_cTVJ · 2022-08-23

**Recommendation:** Accept
**Confidence:** Less certain

**Metareview:**

This paper presents an algorithm for representation and skill learning in RL, based on a contrastive approach. Experiments combining this method with different intrinsic rewards are presented, along with a number of sensitivity analyses.

Overall, the reviewers recognize that there are novel components here despite the work being an extension of existing work. There is some disagreement with regards to the value of these novel components, in particular whether the specific discriminator used is sufficiently evaluated, and whether the proposed method is well-motivated (i.e. by theory). While I am sensitive to these concerns, I find that the paper does a reasonable job at detailing the proposed agent architecture and provides some fresh insights into unsupervised RL.

**Award:**

No

---

### Decision · Program_Chairs · 2022-09-14

Accept